# Teachers during the COVID-19 Era: The Mediation Role Played by Mentalizing Ability on the Relationship between Depressive Symptoms, Anxious Trait, and Job Burnout

**DOI:** 10.3390/ijerph20010859

**Published:** 2023-01-03

**Authors:** Annalisa Levante, Serena Petrocchi, Federica Bianco, Ilaria Castelli, Flavia Lecciso

**Affiliations:** 1Department of Human and Social Sciences, University of Salento, 73100 Lecce, Italy; 2Lab of Applied Psychology, Department of Human and Social Sciences, University of Salento, 73100 Lecce, Italy; 3Faculty of Biomedical Sciences, Università della Svizzera Italiana, 6900 Lugano, Switzerland; 4Department of Human and Social Sciences, University of Bergamo, 24100 Bergamo, Italy

**Keywords:** depression, anxiety, mentalizing, burnout, mediation, teacher, COVID-19, remote teaching

## Abstract

Background: The COVID-19 outbreak caused severe changes in school activities over the past two years. Teachers underwent a re-planning of their teaching approaches, shifting from face-to-face teaching formats to remote ones. These challenges resulted in high levels of burnout. The identification of risk/protective factors contributing to burnout is crucial in order to inform intervention programs. Thus, we hypothesized a mediation role of teachers’ mentalizing ability (processing of emotions, a component of mentalized affectivity) on the relationship between depression, anxiety, and depersonalization (burnout dimension). Two reverse models were computed. Job satisfaction, teachers’ age and gender, school grade, and length of teaching experience served as covariates. Methods: 466 (M(sd) = 46.2 (10.4) years) online questionnaires were completed by Italian teachers of primary (n = 204) and middle (n = 242) schools. Measures of burnout, depression, anxiety, and mentalization were administered. Results: The findings corroborated our hypotheses: in all models, processing emotions served as a mediator on the relationship between depression, anxiety, and depersonalization, and on the reciprocal one. Job satisfaction positively impacted processing emotion, and negatively impacted depression and depersonalization; women teachers reported high levels of the anxious trait. Conclusions: Overall, it can be concluded that the ability to mentalize has a beneficial impact on teachers’ well-being. Policymaking, clinical, and research implications were discussed.

## 1. Introduction

During the first wave of the COVID-19 pandemic in 2020, strict measures of lockdown imposed, among others, a remote format of teaching in many countries [1,2], causing an unprecedented change in school-related activities [3]. During the subsequent year, the uncertain evolution of the pandemic with sudden changes in the numbers of contagion led to the adoption of specific measures of containment. For example, in Italy, primary and middle schools adopted a hybrid teaching format according to the number of positive-tested students in each class. As a sequela of this, every class had to shift from a face-to-face teaching format quickly and unpredictably to a remote one, contributing to create a turbulent working environment for teachers.

The sudden and unstable changes in the teaching-related formats put a strain on educational settings as well as on teachers’ functioning [4] and on their job demands [5]. In this vein, several studies [6,7,8,9,10] investigated the impact of those changes on teachers, showing that they had a high risk to develop stress-related symptomatology, specifically in terms of depressive symptoms and anxiety. The challenges imposed by the management of teaching activities during the COVID-19 pandemic, and the associated stress, had an impact on job burnout. The current paper aimed to investigate teachers’ burnout during the second wave of the COVID-19 pandemic and the role served by several risk and protective factors.

### 1.1. Burnout in Educational Settings

Previous works [11,12] reported that, even in ordinary job scenarios, teachers showed higher levels of stress-related psychological symptoms in terms of depressive thoughts, anxiety, and emotional drain compared with other populations. Consequently, evidence reported an increased risk of mental illnesses and low levels of well-being [13], low job satisfaction [14,15], poor teaching performance [10,16,17], and high risk of job burnout [18].

Burnout is a well-known psychological construct developed by Maslach and colleagues [19], and is defined as an individual’s answer to protracted emotional and interpersonal stressors in a job setting. This construct includes three components: emotional exhaustion (i.e., the physical condition of exhaustion characterized by low energy and chronic fatigue), depersonalization (i.e., cynical attitudes and negative feelings about colleagues), and reduced personal and professional accomplishment (i.e., a sense of disinvestment and of personal and professional failure). Nevertheless, according to Schaufeli and Salanova’s theory regarding job burnout [20], as also confirmed by others [21,22], emotional exhaustion and depersonalization are the two pivotal dimensions of job burnout.

At a global level, during the COVID-19 outbreak, several studies showed that teachers’ job burnout was related to emotional dysregulation and negative emotions [6] affecting the understanding of other mental states, and the use of the remote format to teach and interact with students [8]. Regarding the Italian context, Chirico and colleagues [23] estimated that 16.9% of the teachers surveyed suffered from some forms of burnout and, in particular, teachers working in science, technology, engineering, and mathematics showed higher levels of depersonalization compared with their colleagues teaching humanities subjects. The study by Pellerone [24] confirmed that burnout increased during the lockdown compared with the period before the pandemic. Another study by Bianchi and Caso [25] revealed that the anxiety associated with the use of information and communications technologies, in combination with other stressors, was positively related with job burnout, which, in turn, negatively affected teachers’ well-being. Finally, the study by Procentese and colleagues [9] found that job burnout increased teachers’ stress levels and negatively affected their job satisfaction. Finally, other studies conducted during the second wave of the COVID-19 pandemic revealed that stress-related symptomatology (i.e., anxiety and stress) was positively associated with job burnout [25].

For the specific aims of our study, we decided to devote attention to the depersonalization dimension. Two reasons sustained the proposed research. The depersonalization dimension describes an individual’s mental distancing toward colleagues and/or their job setting [26]. The specificity of the job scenarios during the COVID-19 pandemic exposed teachers to higher levels of burnout and, specifically, to depersonalization [23]. This urges investigators to understand and determine the possible risks and protective factors. Moreover, as claimed by Maslach and Leiter [27], depersonalization leads teachers to leave their jobs; hence, in order to prevent resignations, further investigations on risk and protective factors may provide insights to contribute to subsequent preventive programmes. In accordance with these reasons, the current paper tested the role served by depression symptoms and anxious trait, as risk factors, and mentalizing affectivity, as a protective factor, on teachers’ depersonalization levels.

### 1.2. Depression Symptoms and Anxious Trait as Risk Factors for Depersonalization

Teaching is included among the group of professions with the highest stress-related symptomatology levels [11,28]. Several studies in the pre-COVID-19 era suggested a significant relationship between burnout syndrome, depression, and anxiety [29,30,31]. A meta-analysis [32] established a significant association between anxiety and burnout (r = 0.046) and between depression and burnout as well (r = 0.49). Additionally, a higher prevalence of depressive symptoms was observed in persons with burnout syndrome, contributing to the onset of depressive disorders [33].

During the COVID-19 pandemic, several studies analyzed the relationships between stress-related symptoms, i.e., depression symptoms and anxious trait, and job burnout. Overall, the continuing shift between in-person and remote teaching had a negative impact on teachers’ anxiety and depression levels [34]. Specifically, two studies [35,36] focused on the associations between depression symptoms, anxiety, and job burnout. To be specific, a first study by Cortes-Alvarez and colleagues [37] revealed that levels of depersonalization increased during the pandemic as compared with the pre-pandemic period as well as, as expected, anxiety and depressive symptoms. Those authors also found some positive cross-sectional associations between stress-related symptomatology and job burnout considering two time points: one during the first wave (June 2020) and the second one during the second wave of the COVID-19 pandemic in Mexico (June 2021).

The study by Karakose and colleagues [38] tested a model of relationships in which burnout served as the mediator in the relationship between distress due to COVID-19 and depression. Those authors found that the burnout was determined by distress and was positively associated with depressive symptoms. Those results complemented what had been already found in the literature in the pre-COVID-19 period [39,40,41], demonstrating the existence of a bidirectional relationship between stress-related symptomatology and job burnout.

According to the abovementioned literature, almost all the studies analyzed burnout syndrome as a comprehensive concept and did not consider the specific role of the depersonalization dimension. Therefore, considering the relationships found in previous studies [35,36,37,38] as well as the characteristics of depersonalization (in terms of cynical attitudes and negative feelings), and the role played by depressive symptoms (in terms of absence of positive emotions) and anxiety (in terms of fear), the main purpose of this paper was to consider the inter-relationships between those constructs with four models, which are defined in the dedicated hypotheses section.

### 1.3. Mentalized Affectivity as a Protective Factor for Depersonalization

Mentalization is defined as a human being’s ability to perceive and interpret their own mental states and those of others, such as feelings, desires, thoughts, and beliefs [40]. A good mentalizing ability leads to a coherent understanding of the state of mind and behaviors of both the self and others, whereas a deficit in mentalizing leads to a poor use or integration of mental state information [41,42]. The construct of “mentalized affectivity” proposed by Jurist [43] is a recent enrichment of this subject, which integrates mentalization ability in the process of emotional regulation. This construct strictly connects mentalizing with emotional competence, referring to the adult ability to make sense of one’s own affective experience by performing a reflection on it. Mentalized affectivity includes three components: identifying emotions, expressing emotions, and processing emotions [43,44]. Identifying emotions is the ability to recognize and to label an emotion when it occurs; expressing emotions is the capacity to experience one’s own emotions and communicate them considering one’s own’ mental states and those of others; finally, processing emotions means managing emotions, modulating their intensity, or adapting them according to social settings.

According to the job demands–resource transactional model [45,46], the balance between job demands and job resources permits individuals to reduce their stress and stress-related psychological symptoms as well as to reach job task and goals. Among job resources, Taris and colleagues [45] underlined that a worker’s personal resources should be considered and measured. The personal resources that an individual puts into their job environment are functional to reduce their job demands, stimulate their growth and wellbeing, and to achieve their expected goals [46,47,48]. Among the possible personal resources that people have, self-efficacy and optimism have been studied [49]. In educational settings, depersonalization refers to teachers’ mental distancing toward their students; considering the pivotal role played by teachers’ emotional regulation capabilities to experience less burnout [50], the ability of processing of emotions, as a component of the mentalizing affectivity, may represent an individual resource that allows teachers to manage their negative emotions, modulating their intensity or adapting them according to the work settings. This is what we propose in the subsequent hypotheses, numbered 3 and 4.

### 1.4. Hypotheses

According to the literature review described above on depressive and anxious traits as risk factors for depersonalization, the following hypothesis was formulated:

**H1**.
*It was expected to find a direct impact of teachers’ depression symptoms (Model 1) and their anxious trait (Model 2) on the depersonalization dimension: the higher levels of depression symptoms and anxious trait, the more negative feelings experienced by teachers toward their students/job activities. See Figure 1.*


Since the study had a cross-sectional design and research has demonstrated bidirectional relationships between stress-related symptomatology and job burnout [37,38,39], a further backward hypothesis was proposed as follows:

**H2**.
*To find a direct effect of the depersonalization on depression symptoms (Model 3) and anxious trait (Model 4): the lower negative feelings experienced by teachers toward their students/job activities, the lower their levels of depressive symptoms and anxious traits. See Figure 2.*


The present study also considered a protective mediation mechanism in terms of the ability to process emotions on depersonalization and explored its impact via two further hypotheses:

**H3**.
*It was expected to find a mediation effect of the teachers’ mentalized affectivity ability on the relationship between depressive symptoms (Model 1) and anxious trait (Model 2), and depersonalization: the lower levels of stress-related symptomatology, the lower the level of negative feelings experienced by teachers toward their students/job activities via the mediation role served by higher levels of the teachers’ emotion processing ability. See Figure 1.*


**H4**.
*It was expected to find a mediation effect of the mentalized affectivity ability, in terms of processing of emotions, on the relationship between depersonalization, depressive symptoms (Model 3), and anxious trait (Model 4): the lower the level of negative feelings experienced by teachers toward their students/job activities, the lower the levels of depression symptoms and anxious trait via the mediation role served by higher levels of teachers’ emotion processing ability. See Figure 2.*


## 2. Materials and Methods

### 2.1. Procedure

Cross-sectional data collection was performed between September 2021 and January 2022 during the second wave of the COVID-19 pandemic. At that time, in Italy, primary and middle schools adopted a hybrid teaching format according to the number of positive-tested students in each class. Consequently, classes shifted from a face-to-face teaching format quickly and unpredictably to a remote one, contributing to create a turbulent working/learning environment for teachers and their students.

The questionnaire was imported on Microsoft Form and the link was spread via mainstream social platforms (Whatsapp, Bergamo, Italy) and via an internal communication provided by the local school office. The school office is a local and peripherical office of the Italian Ministry of the Education responsible for monitoring the schools in each district and promoting rules and regulations. The local school office sent out a communication to schools and teachers with the invitation to participate into this research. The local school office itself did not take part in the research.

The questionnaire did not require any participants’ registration. Only one inclusion criterion was pre-defined, i.e., being an Italian teacher of primary or middle school. The rationale underlying this choice considered that primary and middle school grades were the most affected by the sudden and unpredictable changes during the second wave of the COVID-19 pandemic, making them a good case study for our research. The presentation order of the measures follows the order of the measures in the subsequent section.

The study was conducted in compliance with the Italian rules and regulations for research in psychology, with the ethical requirements for research in psychology, and according to the GDPR (General Data Protection Regulation), which regulates data protection in Europe. Participation in the survey was voluntary and participants gave their informed e-consent (i.e., online informed consent) prior to answering the questions. They were informed about the purposes of the research, the modality of the data collection, and about their rights. They also knew that they could withdraw from the study anytime. Participants did not receive any compensation for their participation.

The Ethical Committee of the University of Salento gave its approval (no. 71084; 5 May 2021).

### 2.2. Participants

Four-hundred and sixty-six participants completed the questionnaire, with an average age of 46.2 years (SD = 10.4 years; age range = 22–68 years; women: 388; 87%). Their educational levels were intermediate (up to 13 years) for 106 (3.8%), high (up to 18 years) for 245 (55%), and 95 (21.3%) of them reached post lauream (Ph.D. degree) levels. Regarding occupational status, 62 (13.9%) teachers had a permanent contract, 56 (12.6%) were special-needs teachers, and 328 (73.5%) teachers had a temporary-employment contract. Regarding the school grade, 204 (45.7%) were teachers of primary school and 242 (54.3%) were teachers of middle schools. Two-hundred and ninety-four (65.9%) of the teachers were in a stable relationship. In addition, 274 (61.4%) had at least one child, whereas the remaining 172 (38.6%) had no children. Finally, the average length of their teaching experience was 16.05 years (SD = 11.76 years; range = 1–41 years).

### 2.3. Measures

Depressive Symptoms and Anxious Trait. Fourteen items from the Depression Anxiety Stress Scale (DASS-21; [51]) were used to evaluate teachers’ depressive symptoms and anxious trait. According to Bottesi and colleagues [51], depressive symptoms include an individual’s lack of incentive and low self-esteem, and anxious trait refers to an individual’s somatic responses to acute fear. Respondents were asked to rate the frequency of some behaviors on a 4-point Likert scale ranging from 0 (It has never happened) to 3 (It always happens). Two examples include the following: “I felt life was meaningless” was used in order to evaluate depressive symptoms, and “I felt I was close to panic” was used in order to evaluate anxiety. Two partial scores as the average of items were computed, and higher scores indicated higher depressive and anxiety levels.

Processing of Emotions. Seven items from the Mentalized Affectivity Scale (MAS; [52]) were administered to assess teachers’ mentalizing affectivity in terms of emotional processing ability, that is, an individual’s ability to manage emotions in terms of modifying and/or refining them in a new experience. Respondents were asked to rate their degree of agreement with a set of statements on a 7-point Likert scale ranging from 1 (Strongly disagree) to 7 (Strongly agree). An example of an item is the following one: “When I am filled with negative emotion, I know how to handle it”. A total score as the average of the items was calculated and higher scores indicated a higher ability to process emotions.

Depersonalization. Five items from the Maslach Burnout Inventory (MBI; [53]) were used to measure teachers’ depersonalization, that is, the negative attitude and feelings toward their own work and the increased mental distance toward their students/job activities. Respondents were asked to rate the frequency of effects on a 5-point Likert scale ranging from 1 (Never) to 5 (Every day). Two examples include the following ones: “I seem to treat some students as if they were impersonal objects” and “I am afraid that this work can harden me emotionally”. The total score as the average of items was computed and higher scores indicated higher negative attitudes toward one’s own work and an increased psychological distance toward one’s own students.

Covariates. For job satisfaction, one ad hoc item was defined (“How much are you satisfied with your job?”) with response options ranging from 1 (Not at all) to 10 (Very much). In addition, the primary vs. middle school grades, the teachers’ age and gender, and the number of years of their teaching experience were included.

### 2.4. Statistical Analyses

The statistical analyses were performed using SPSS version 25 [54]. No missing data imputation techniques were performed because the items had a forced-choice requirement. To perform group (gender and school grades) comparisons, parametric (Independent sample *t*-test) and nonparametric (Mann–Whitney U) tests were performed. To evaluate the associations between variables, Pearson’s rho correlations were carried out. Mediations were performed using Process v3.0, applying Model 4 and 5000 bootstraps inference for model coefficients. In the mediation models, teacher job satisfaction, age and gender, primary vs. middle school grade, and the years of job experience were included as covariates.

## 3. Results

### 3.1. Preliminary Results

Table 1 shows the descriptive data regarding the measures of depersonalization, depression symptoms and anxious trait, and processing of emotions for both primary vs. middle school grade teachers and according to gender.

### 3.2. Comparison across Teacher’ Gender and School Grades

Table 2 shows the mean scores, standard deviations, and statistical coefficients. The findings showed that women teachers reported significantly higher scores on stress-related symptomatology, in terms of depression symptoms and anxious trait, than men teachers.

In addition, we compared the degree of depersonalization, the levels of depression symptoms and anxious trait, and the ability to process emotions across primary vs. middle school grades. Table 3 reports the mean scores, standard deviations, and statistical coefficients. The findings showed no significant differences between the two groups.

### 3.3. Correlations between Measures for Each Group of Teachers

Regarding the sub-sample of participants teaching primary school, the degree of depersonalization was positively and significantly associated with depressive symptoms and anxious trait, indicating that the higher the mental distance, the higher the stress-related symptomatology experienced by participants. Additionally, the degree of depersonalization was negatively associated with job satisfaction: in other words, the more negative feelings toward their students/job activities, the lower satisfaction teachers perceived with their jobs. Table 4 shows the correlations between the measures for each group of teachers. 

As well as the sub-sample of teachers of primary school, the degree of depersonalization of those who teach in middle school was positively associated with depression and anxious trait; in addition, the burnout dimension (i.e., depersonalization) was negatively associated with teachers’ mentalizing affectivity ability, in terms of processing of emotions, and job satisfaction, indicating that the more negative feelings toward students/job activities, the lower the ability to modulate intensity or adapting one’s own feelings to the educational setting, and the lower the job satisfaction. Teachers’ age and their teaching experience in terms of seniority were only reciprocally associated.

### 3.4. Mediation Models

The hypothesized Models 1 and 2 shown in Figure 1 were tested with job satisfaction, teacher age and gender, primary vs. middle school grade, and the years of teaching experience as covariates. Figure 3 and Figure 4 show the results of Models 1 and 2, respectively.

For both the estimated mediation models, the total (Model 1: F_(5,391)_ = 12.279; *p* < 0.001 and Model 2: F_(5,391)_ = 13.618; *p* < 0.001) and indirect (Model 1: β: 0.028; BootLLCI = 0.000; BootULCI = 0.062; Model 2: β: 0.023; BootLLCI = 0.001; BootULCI = 0.053) effects were significant. In other words, depressive symptoms (Model 1) and anxious trait (Model 2) reported by teachers were related to their degree of depersonalization toward their students/job activities via the mediation of the their ability to process and/or modulate their own emotions. In sum, HP1 and HP2 were corroborated: the lower levels of depressive symptoms and anxious trait, the lower the level of negative feelings experienced by teachers toward their students/job activities via the mediation role served by higher levels of emotional processing ability. In addition, in Models 1 and 2 job satisfaction was the only covariate significantly associated with the outcome.

The two reverse mediation Models 3 and 4 are shown in Figure 5 and Figure 6, respectively. For both mediation models estimated, the total (Model 3: F_(5,391)_ = 15.730; *p* < 0.001 and Model 4: F_(5,391)_ = 10.208; *p* < 0.001) and indirect (Model 3: β: 0.037; BootLLCI = 0.010; BootULCI = 0.071; Model 4: β: 0.031; BootLLCI = 0.006; BootULCI = 0.065) effects were significant. In other words, the degree of depersonalization reported by teachers were related to their depression symptoms (Model 3) and anxious trait (Model 4) via the mediation role played by their ability to process and/or modulate their emotions. Namely, the results supported HP3 and HP4: the lower negative feelings experienced by teachers toward their students/job activities, the lower the levels of depressive symptoms and anxious trait via the mediation role served by higher levels of emotional processing ability. In addition, in Model 4 the teachers’ gender was the only covariate significantly associated with the outcome: this means that women teachers showed higher levels of anxiety than men teachers.

## 4. Discussion

The impact of the COVID-19 pandemic has been widely investigated in healthcare professionals [55,56], parents of typically [42,57,58,59] and atypically [60,61,62] developing children, and teachers [7,23,25,63]. The present study places itself in this research field, investigating the impact of the challenging scenario caused by the COVID-19 pandemic on teachers working in primary and middle schools. During the COVID-19 emergency, the “new normal” job setting [36], characterized by a considerable use of technologies and by the definition of new teaching strategies to involve students during lessons, has directly and heavily influenced teachers’ stress [64] and burnout [65,66,67]. Specifically, the identification of risk factors as well as of protective ones involved in the explication of job burnout might then be important to support teachers in the future and have created a case study for the present research. Indeed, the main purpose of the current study was to examine the interplay between the teachers’ job burnout, their stress-related symptomatology (i.e., as risk factors), and their mentalizing affectivity (i.e., as a protective factor) though mediation models. To be specific, we hypothesized that the stress-related symptomatology, in terms of depressive symptoms (Model 1) and anxious trait (Model 2), impacted teachers’ depersonalization (HP1). We hypothesized backwards relationships between those constructs (see Model 3 and Model 4; HP2). We also expected that the relationships between those constructs would be mediated by the mentalizing affectivity, in terms of emotional processing ability (see Models 1–3, HP3 and HP4).

In sum, our results revealed a positive direct path between stress-related symptomatology and depersonalization, demonstrating HP1: that is, the higher levels of depression and anxiety, the more negative feelings were experienced by teachers toward their students/job activities. Afterward, we tested the reverse mediation models, where teachers’ depersonalization impacted their mentalizing affectivity, which, in turn, influenced depressive symptoms (Model 3) and anxious trait (Model 4). This was due to the fact that literature underlined bi-directional relationships between the considered variables and this study had a cross-sectional study design. Again, our results corroborated those hypotheses as well, revealing that (a) more negative emotions and a higher sense of alienation experienced by teachers were positively correlated with higher depression and anxiety levels (HP2).

Overall, the evidence of this study suggests that, on one side, during the COVID-19 outbreak, the negative feelings and emotions experienced by teachers set the stage for a sense of alienation [67]. Teachers experienced an overwhelming environment and worked under the presence of negative emotions and worries, constantly searching for new and appropriate strategies to provide adequate support to their students [68]. Therefore, the specificity of the job scenarios during the COVID-19 pandemic exposed teachers to higher levels of stress-related symptomatology to which they responded by mentally distancing themselves from their students and job demands, probably as a sort of self-protective mechanism. On other side, the cynicism, dehumanization, and negative emotions characterizing teachers’ depersonalization may be related to higher depression and anxiety, supporting the hypothesis that job burnout is a phase in the development of stress-related symptomatology [69,70,71]. Of course, further studies to test this explanation are required. Anyway, one main consideration here is that our evidence suggested that research should not only examine the overall concept of burnout, but also the more specific depersonalization dimension because it seems it has been affected by the COVID-19 pandemic as well.

Our study also confirmed the indirect relationships between depression, anxiety, and depersonalization, via the mediation of the mentalizing affectivity. Indeed, our third hypothesis (HP3) and fourth hypothesis (HP4) were confirmed: the lower levels of depression and anxiety, the lower sense of alienation (i.e., depersonalization), via the mediation of higher mentalizing affectivity skills (i.e., processing emotions), and vice versa. Although still preliminary, our results support the idea that the teachers’ ability to process emotions, to modify and adapt them according to the school setting, represents a protective factor. Mentalizing affectivity represents a protective factor against the development of stress-related symptomatology, as demonstrated by others [42,72,73]. Good mentalizing affectivity allows individuals to respond to negative experiences, feelings, and emotions, and to process them adequately, reducing the negative impact of those experiences on wellbeing [73]. In addition, our study showed some bidirectional and circular relationships among the considered variables that future longitudinal investigations should deepen by applying more sophisticated design and statistical models, such as the random-intercept cross lag model, which is able to identify the inter-relationships among variables over time. Therefore, this ability could act as a personal resource to face job demands, as anticipated by the job-demands model [46]. Moreover, our results are consistent with some evidence [72,74] suggesting that mentalizing predicts individual’s wellness [74], decreasing stress-related affective arousal.

As regards the covariates included in our models, only teachers’ job satisfaction had a significant effect on the outcome. Job satisfaction is an attitude that people form towards their job by considering their feelings, beliefs, and behaviors [75]. Our results highlighted that job satisfaction negatively affected teachers’ depersonalization, as previously demonstrated by Platsidou [76] in the pre-pandemic era.

Finally, as a complement of the main analyses, our results highlighted that women teachers reported higher stress-related symptomatology (i.e., anxious trait) than men. Albeit this result should be interpreted cautiously because of the unbalanced gender distribution in our sample, previous studies carried out during the pandemic [77] and in the pre-pandemic era [78] confirmed that women teachers reported higher levels of stress than their men counterparts. This set of evidence needs to be carefully considered in future studies, because it reveals that in the school context women are at significantly greater risk of suffering from stress-related symptomatology than men.

## 5. Limitations, Conclusions, and Implications

The results of the present study must be considered in light of several limitations. The first one is related to the cross-sectional nature of the study: hence, causal relationships between the considered variables cannot be drawn. Longitudinal studies are needed to expand the knowledge regarding the nature and the directions of the paths tested in the current study. Second, our sample included mainly women teachers. Although, teachers in Italy are mostly women, 95.6% and 77% in primary and middle schools respectively (https://ec.europa.eu/eurostat/databrowser/view/educ_uoe_perd03/default/table?lang=en, accessed on 20 November 2022), and our sample represents the real situation quite well, further studies should better stratify the sample according to gender, and perhaps school grade, to shed light on the possible under-studied phenomenon of stress on men teachers.

Despite the abovementioned limitations, our results suggest two possible routes for research and intervention. Depersonalization appears to be particularly affected by the COVID-19 pandemic. Depersonalization is not only one of the most important dimensions of the burnout, having an impact on the teachers’ functioning, wellbeing, job adaptation, and emotions, but could have also a significant and negative impact on students’ wellbeing because it directly implies that feelings are detached and be mentally distant from the recipients of the job, which are the students themselves. Therefore, our results encourage more attention to the dimension of depersonalization in terms of the double-downsides it has on both the teachers and the students.

Our evidence also suggests that the ability to mentalize has a positive and beneficial impact on teachers’ well-being, in terms of low levels of depersonalization, depressive symptoms, and anxious trait. This evidence proposes some clinical and practical implications. It may be relevant to remember that teachers, as well as other professionals employed in healthcare settings [79,80,81], play a pivotal role from an educational, affective, mentalistic, and relational point of view and represent a resource for their students during difficult situations [82,83]. Hence, their mental health should be closely monitored to prevent the onset of mental illness and/or the worsening of their well-being, leading to job burnout and/or resignation. In this vein, mentalized-based intervention programs for teachers could positively affect their job setting, in terms of lowering the levels of negative emotions and the sense of alienation toward students/job activities. From a policy maker point of view, institutions and educational systems should join their efforts in providing teachers intervention programs based on self-care strategies and on the development of personal resources, which we demonstrated to be protective factors during stressful events. For example, it could be useful to transform schools in mentalizing organizations [83]: in this vein, a so called ‘mentalizing school’ may strengthen individual–environment bonds by creating a care setting, improving the members’ curiosity toward each other’s thoughts and feelings [83,84], and also facilitating the ability to think, symbolize, and mentalize emotions. These intervention programs could be designed via the collaboration of professionals in the fields of educational and work/organizational settings. This would lead to a less probable alienation and depersonalization effect of stressful job environments.

## Figures and Tables

**Figure 1 ijerph-20-00859-f001:**
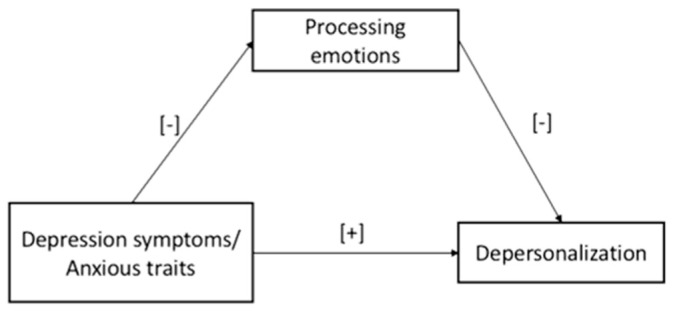
The hypothesized Models 1 and 2.

**Figure 2 ijerph-20-00859-f002:**
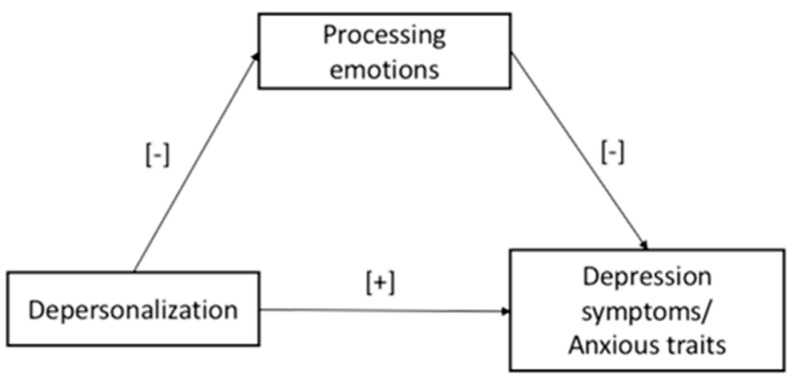
The hypothesized Models 3 and 4.

**Figure 3 ijerph-20-00859-f003:**
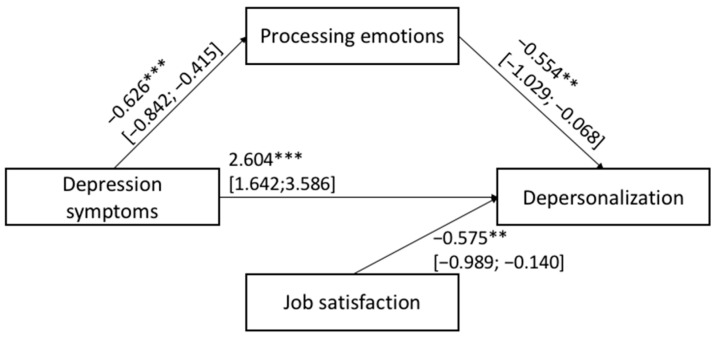
Results of mediation: Model 1. Significance levels were *** *p* < 0.001; ** *p* < 0.010; the bootstrap confidence intervals are reported in parentheses, while non-significant covariates are not reported in the figures.

**Figure 4 ijerph-20-00859-f004:**
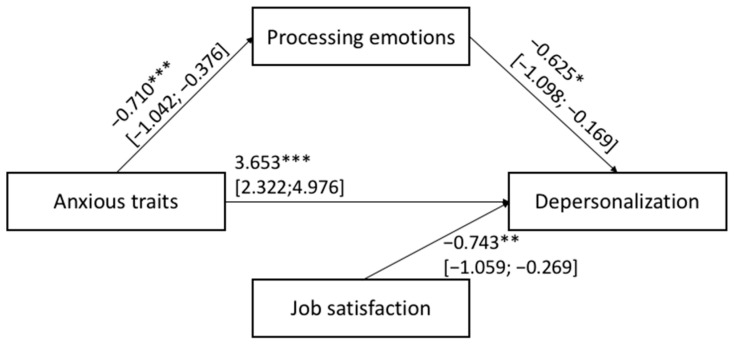
Results of mediation: Model 2. Significance levels were *** *p* < 0.001; ** *p* < 0.010; * *p* < 0.050; the bootstrap confidence intervals are reported in parentheses, while non-significant covariates are not reported in the figures.

**Figure 5 ijerph-20-00859-f005:**
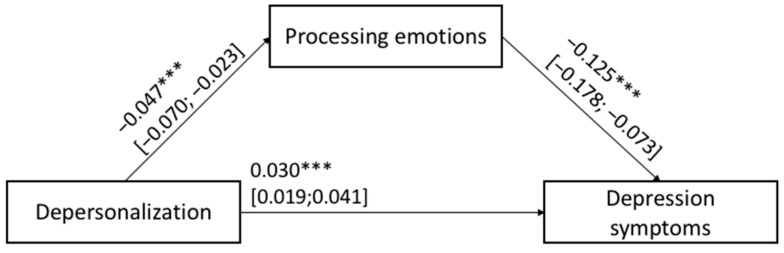
Results of mediation: Model 3. Significance levels were *** *p* < 0.001; the bootstrap confidence intervals are reported in parentheses, while non-significant covariates are not reported in the figures.

**Figure 6 ijerph-20-00859-f006:**
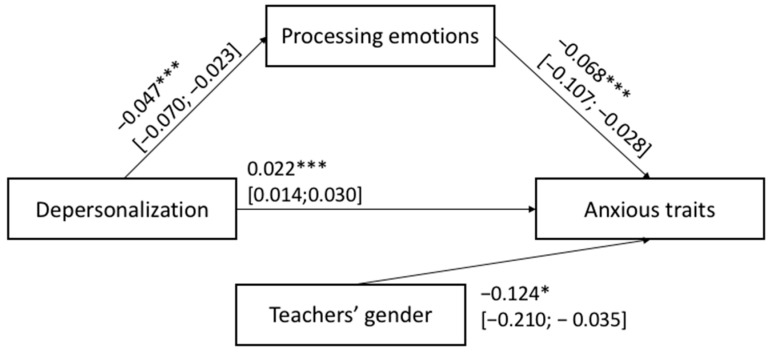
Results of mediation: Model 4. Significance levels were *** *p* < 0.001; * *p* < 0.050; the bootstrap confidence intervals are reported in parentheses, while non-significant covariates are not reported in the figures.

**Table 1 ijerph-20-00859-t001:** Mean scores and standard deviations of measures of depersonalization, depression symptoms, anxious trait, and processing of emotions for each group of teachers and for gender.

School Grade	Gender		M(SD)
Teachers working in a primary school	Women	Depersonalization (Range = 0–30)	9.18 (5.36)
		Depression symptoms (Range = 1–4)	1.57 (0.53)
		Anxious trait (Range = 1–4)	1.41 (0.40)
		Processing of Emotions (Range = 1–7)	4.61 (1.05)
	Men	Depersonalization (Range = 0–30)	7.12 (3.83)
		Depression symptoms (Range = 1–4)	1.21 (0.15)
		Anxious trait (Range = 1–4)	1.18 (0.15)
		Processing of Emotions (Range = 1–7)	4.45 (1.52)
Teachers working in a middle school	Women	Depersonalization (Range = 0–30)	8.79 (4.91)
		Depression symptoms (Range = 1–4)	1.54 (0.55)
		Anxious trait (Range = 1–4)	1.39 (0.40)
		Processing of Emotions ( Range = 1–7)	4.59 (1.16)
	Men	Depersonalization (Range = 0–30)	7.52 (4.16)
		Depression symptoms (Range = 1–4)	1.46 (0.53)
		Anxious trait (Range = 1–4)	1.23 (0.27)
		Processing of Emotions (Range = 1–7)	4.87 (0.83)

**Table 2 ijerph-20-00859-t002:** Comparison on measures of depersonalization, depression symptoms, anxious trait, and processing of emotions across teachers’ gender.

	Women (n = 388) M(SD)	Men (n = 78) M(SD)	Mann–Whitney U; *p*-Value
Depersonalization	8.99 (5.14)	7.46 (4.09)	U = 9.499; *p* = 0.055
Depression symptoms	1.56 (0.54)	1.42 (0.51)	U = 8.988,5; *p* = 0.013
Anxious trait	1.4 (0.4)	1.22 (0.25)	U = 8.323,5; *p* = 0.001
Processing emotions	4.6 (1.11)	4.81 (0.95)	U = 10.140,5; *p* = 0.224

**Table 3 ijerph-20-00859-t003:** Comparison of measures of depersonalization, depression symptoms, anxious trait, and processing of emotions across school grades.

	Primary School (n = 204) M(SD)	Middle School (n = 242) M(SD)	*t*-Test; *p*-Value
Depersonalization	9.1 (5.32)	8.53 (4.79)	*t*(444) = 1.188; *p* > 0.05
Depression symptoms	1.56 (0.53)	1.52 (0.55)	*t*(444) = 0.703; *p* > 0.05
Anxious trait	1.4 (0.4)	1.35 (0.38)	*t*(444) = 1.186; *p* > 0.05
Processing emotions	4.6 (1.07)	4.65 (1.1)	*t*(444) = 0.423; *p* > 0.05

**Table 4 ijerph-20-00859-t004:** Correlations between measures for each group of teachers enrolled in the study.

	Depersonalization	(1)	(2)	(3)	(4)	(5)	(6)
Depersonalization	-	0.379 ***	0.377 ***	−0.090	−0.173 **	0.015	−0.021
Depression symptoms (1)	0.277 ***	-	0.697 ***	−0.245 ***	−0.218 **	−0.046	−0.089
Anxious trait (2)	0.285 ***	0.629 ***	-	−0.219 **	−0.065	−0.035	−0.058
Processing emotions (3)	−0.325 ***	−0.390 ***	−0.312***	-	0.185 **	−0.017	−0.043
Job satisfaction (4)	−0.355 ***	−0.389 ***	−0.275 ***	0.281 ***	-	−0.027	−0.052
Teachers’ age (5)	−0.025	−0.159 *	−0.040	0.157 *	0.058	-	0.777 ***
Length of teaching experince (6)	−0.046	−0.112	−0.002	0.126	0.124	0.773 ***	-

Note: *** *p* < 0.001; ** *p* < 0.010; * *p* < 0.050. We reported above the longest diagonal (marked with a dash) the correlations regarding teachers of primary school; under the longest diagonal we reported the coefficients regarding the correlations of the middle school teachers group.

## Data Availability

Data are available to the first author upon reasonable request.

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
