# Peer review of "Teachers during the COVID-19 Era: The Mediation Role Played by Mentalizing Ability on the Relationship between Depressive Symptoms, Anxious Trait, and Job Burnout"

_ijerph, 2023, doi:10.3390/ijerph20010859_

Round 1

Reviewer 1 Report

This is a well-organized and well-written article. The main research is teachers' mental shaping ability on the relationship between repressive symptoms, anxious trait, and job burnout. But I have to say that there are still many places in the article that need to be revised, as follows:

First, about the introduction. Putting the research background and research hypothesis together in the introduction leads to insufficient discussion of key points. The research background and value embodiment are unclear, and the difference and contribution from the existing research are insufficient. The discussion of existing research and the proposal of research hypothesis are too simple. It is suggested that the authors divide the introduction, literature review and hypothesis into two parts. At the same time, it should be noted that the theoretical basis of this study needs to be clearly explained, and the existing statements are not enough to support this research hypothesis.

Second, about research methods. Why is the explanation of the sample at the end, explaining the research process first? This is not in line with the general reporting logic. At the same time, the authors need to clearly explain the research process and ensure the operation of ethical norms.

Third, about the result. A, the analysis of descriptive statistics is too thin, or it doesn't make any sense. The authors almost finished the analysis results in one sentence. What is the significance? B, there are many unknowns in correlation analysis, and the authors need to explain them. C, since this data is cross-sectional data, it is suggested that the authors add common method deviation test.

Fourthly, about discussion and results. Lack of exposition of educational critical thinking, or reflection from research results.

All in all, it's a good manuscript, and there are still many details to proofread except the obvious points mentioned above. I hope the authors can treat it strictly, and I look forward to their revised articles.

Author Response

#1

Comments and Suggestions for Authors

This is a well-organized and well-written article. The main research is teachers' mental shaping ability on the relationship between repressive symptoms, anxious trait, and job burnout. But I have to say that there are still many places in the article that need to be revised, as follows:

Authors: we are very grateful for the revision you have made to our paper. based on your suggestions and considerations, the paper changed substantially and improved a lot. we think the present version has been much better than the first one and could give a good contribution to the literature. We have provided our answers under each of your point, and we have changed the paper using track changes. Thank you again for your commitment and work on this paper.

First, about the introduction. Putting the research background and research hypothesis together in the introduction leads to insufficient discussion of key points. The research background and value embodiment are unclear, and the difference and contribution from the existing research are insufficient. The discussion of existing research and the proposal of research hypothesis are too simple. It is suggested that the authors divide the introduction, literature review and hypothesis into two parts. At the same time, it should be noted that the theoretical basis of this study needs to be clearly explained, and the existing statements are not enough to support this research hypothesis.

Authors: we have split the introduction in two parts, one with the background information for the research and the other one with the hypotheses. We have also included more details about the theoretical bases of the underlined hypothesized relationships between variables.

Second, about research methods. Why is the explanation of the sample at the end, explaining the research process first? This is not in line with the general reporting logic. At the same time, the authors need to clearly explain the research process and ensure the operation of ethical norms.

Authors: The materials and Methods paragraph now starts with the procedure and description of the participants. Measures comes third, and finally the statistical analyses. In the procedure section we have included some more details about the procedure and compliance with the ethical norms.

Third, about the result. A, the analysis of descriptive statistics is too thin, or it doesn't make any sense. The authors almost finished the analysis results in one sentence. What is the significance? B, there are many unknowns in correlation analysis, and the authors need to explain them. C, since this data is cross-sectional data, it is suggested that the authors add common method deviation test.

Authors: thank you for pointing out this point. In our paper, the results took 5 pages from line 346 to line 452 (please consider that the figures and tables are not included in the lines count). The results section also includes:

  1. one descriptive table with means and standard deviations of the measures considering the gender and grades of the teachers (it takes almost one entire page);
  2. one table in which female and male teachers have been compared by all the relevant variables: this table includes descriptive statistics (Ms and SDs)
  3. one table with comparisons between primary vs middle school teachers on the relevant variables (the table includes Ms and SDs)
  4. one correlation table with all the relevant statistics, r-values and p-values. The values above the diagonal (marked with a dash) reported the correlations and p-values between variables for the primary school teachers; the values below the diagonal reported the correlations and p-values for the middle school teachers. There are no unknown values in this table, all the values are marked, and all the variables are clearly stated. the table has also a section below called Note with explanation of the p-values * ** and ***, and with the explanation of the above/below diagonal values.
  5. paragraph 3.4 reports the results of the 4 mediation analyses, with F-results and degree of freedoms, 5000 bootstraps inference for model coefficients, intervals of confidence, beta-values, and significance levels. We also included 4 figures with detailed results reported in there.
  6. we are fully aware that the study has a cross-sectional design. for this reason, the relationship between variable have been analyzed in the hypothesized direction, see Model 1 and 2, and in the opposite direction, see Model 3 and 4. We also carried out moderation analyses with bootstrap resampling method for the same reason. Finally, we have acknowledged the fact that our study has a cross-sectional design in the limitation section, and we have analyzed both sides of the story in the discussion section.

We feel that the included results give to the reader a comprehensive picture of the analyses we have done and the evidence we have collected with this study. If you feel that something is missing, we would be happy to consider the possibility to add other specific statistics.

Fourthly, about discussion and results. Lack of exposition of educational critical thinking, or reflection from research results.

Authors: in the discussion section, we claimed that our evidence suggested that not only the overall concept of burnout, but also the more specific depersonalization dimension has been touched by the COVID-19 pandemic both as outcome and predictor variable of the anxiety and depression. Another quite important result is about the protective role played by the mentalizing affectivity, and the processing emotions. We have reorganized the discussion section according to you suggestion and gave some more insight to the discussion. We have included some more specific consideration regarding the educational settings and some critical thinking about the implications of our results. Thank you for pointing out this consideration.

All in all, it's a good manuscript, and there are still many details to proofread except the obvious points mentioned above. I hope the authors can treat it strictly, and I look forward to their revised articles.

Reviewer 2 Report

The article is relevant to the journal's mission. The paper is relevant for several reasons. It is a study that contributes to increasing the field of knowledge in relation to depression, anxiety and burnout, 2. It is of vital importance in providing insight that can contribute to improving the well-being of teachers through mentalizing ability.

The topic of the article "Teachers during the COVID-19 Era: The Mediation Role Played 2 by Mentalizing Ability on the Relationship between Depres-3 sive Symptoms, Anxious Trait, and Job Burnout" is interesting and a timely study, as it constitutes an emerging research problem in relation to the severe changes in school activities brought about by COVID-19.

The title is in line with the content of the article, as is the abstract, which clearly explains the objectives, methodology and results of the study.

The document is well structured, facilitating understanding of the study. The theoretical foundation is based on the research questions and the objective of the study, where the depersonalisation dimension acquires a determining value. Contextualised and current bibliography is provided.

Objective: The research problem and the objective of the study are well defined.

Method: Not all details are specified in the procedure. This evaluator considers that the research design used needs to be made explicit, as it is only referred to in the limitations section.

The research phases are presented in a clear and structured way and the research questions are answered in a clear and detailed manner.

The selection criteria are clearly and explicitly specified. 

Results: This evaluator considers the results shown in terms of the study problem to be relevant, as they reveal how the ability to mentalise has a positive and beneficial impact on the well-being of teachers.

The study clearly specifies the limitations and possible implications for educational practice. 

All in all, I consider this to be an excellent study that will contribute to the advancement of knowledge regarding the influence of certain factors in work environments.

Author Response

Comments and Suggestions for Authors

The article is relevant to the journal's mission. The paper is relevant for several reasons. It is a study that contributes to increasing the field of knowledge in relation to depression, anxiety and burnout, 2. It is of vital importance in providing insight that can contribute to improving the well-being of teachers through mentalizing ability.

The topic of the article "Teachers during the COVID-19 Era: The Mediation Role Played 2 by Mentalizing Ability on the Relationship between Depres-3 sive Symptoms, Anxious Trait, and Job Burnout" is interesting and a timely study, as it constitutes an emerging research problem in relation to the severe changes in school activities brought about by COVID-19.

The title is in line with the content of the article, as is the abstract, which clearly explains the objectives, methodology and results of the study.

The document is well structured, facilitating understanding of the study. The theoretical foundation is based on the research questions and the objective of the study, where the depersonalisation dimension acquires a determining value. Contextualised and current bibliography is provided.

Objective: The research problem and the objective of the study are well defined.

Method: Not all details are specified in the procedure. This evaluator considers that the research design used needs to be made explicit, as it is only referred to in the limitations section.

The research phases are presented in a clear and structured way and the research questions are answered in a clear and detailed manner.

The selection criteria are clearly and explicitly specified. 

Results: This evaluator considers the results shown in terms of the study problem to be relevant, as they reveal how the ability to mentalise has a positive and beneficial impact on the well-being of teachers.

The study clearly specifies the limitations and possible implications for educational practice. 

All in all, I consider this to be an excellent study that will contribute to the advancement of knowledge regarding the influence of certain factors in work environments.

Authors: we are very grateful for the revision you have made to our paper. We have provided much more details about the procedure in this revised version; you can find them on the dedicated section. We have underlined the changes to the paper using track changes. Thank you again for your commitment and work on this paper.

Reviewer 3 Report

1.     This is a meaningful study as this manuscript contributes to furthering the analysis of teachers during the COVID-19 Era.

2.     The title and abstract are appropriate for the content of the text.

3.     The article is well constructed.

4.     The analysis was well performed.

5.     The manuscript is well-written.

6.     The conclusion could be strengthened by adding some objective statements about the contributions of this study as a reference during the COVID-19 Era.

Author Response

Comments and Suggestions for Authors

  1. This is a meaningful study as this manuscript contributes to furthering the analysis of teachers during the COVID-19 Era.
  2. The title and abstract are appropriate for the content of the text.
  3. The article is well constructed.
  4. The analysis was well performed.
  5. The manuscript is well-written.
  6. The conclusion could be strengthened by adding some objective statements about the contributions of this study as a reference during the COVID-19 Era.

Authors: we are very grateful for the revision you have made to our paper. We have revised the paper according to your and the other two reviewers’ comments. We have underlined the changes to the paper using track changes. Thank you again for your commitment and work on this paper.

Round 2

Reviewer 1 Report

Agree with the changes made by the authors. Congratulations on their good research.